# 1 Climate Variabilities Synergistically Influence Marine

## **2** Heatwaves in the North Sea

- Yuxin Lin<sup>1,2</sup>, Zhiqiang Liu<sup>1,3\*</sup>, Feng Zhou<sup>4,5</sup>, Qicheng Meng<sup>4,5</sup>, Wenyan Zhang<sup>2\*</sup>
- Department of Ocean Science and Engineering, Southern University of Science and Technology, Shenzhen, China
- <sup>2</sup>Institute of Coastal Systems—Analysis and Modeling, Helmholtz-Zentrum Hereon, Geesthacht, Germany.
- <sup>3</sup>Center for Complex Flows and Soft Matter Research, Southern University of Science and Technology, Shenzhen,
- China
- 4State Key Laboratory of Satellite Ocean Environment Dynamics, Second Institute of Oceanography, Ministry of
- Natural Resources, Hangzhou, China
- <sup>5</sup>Observation and Research Station of Yangtze River Delta Marine Ecosystems, Ministry of Natural Resources,
- Zhoushan, China

*Correspondence to:* Wenyan Zhang (wenyan.zhang@hereon.de), Zhiqiang Liu (<u>liuzq@sustech.edu.cn</u>)

19

22

27

39

43

47

Abstract

Global shelf seas have experienced unprecedented marine heatwaves (MHWs) in recent decades. Although state-of-the-art forecast systems show skilful prediction of MHWs in tropical regions, their limited performance elsewhere highlights the need for a more complete mechanistic understanding at regional scales. Here, we examine MHWs in the Northeastern Atlantic shelf, a region strongly influenced by multiple climate variabilities. Using a correlation-based k-means clustering approach, we identified two distinct subregions with contrasting seasonal patterns. The southern North Sea (Cluster 1) exhibits increased MHW frequency, intensity, and duration in winter, primarily associated with a positive East Atlantic Pattern that typically follows a negative North Atlantic Oscillation in late autumn. These conditions intensify westerly winds and enhance warm Atlantic inflow through both atmospheric and oceanic pathways. In contrast, the northern North Sea (Cluster 2) shows enhanced MHW frequency and duration in summer, driven by teleconnections across multiple ocean basins. The Atlantic Multidecadal Variability modulates these linkages, with its positive phase strengthening Pacific-Atlantic connections via Rossby wave propagation. This north-south contrast demonstrates that different combinations of atmospheric and oceanic processes shape MHW variability across the shelf, providing a physical basis for improving regional MHW prediction.

#### 1 Introduction

Marine heatwayes (MHWs), defined as anomalous warm seawater events (Pearce et al., 2011), have become more frequent, persistent, and intense over the past decades, with persistent temperature extremes significantly affecting marine ecosystems and commercial fisheries (Yan et al., 2020). These trends are largely attributed to anthropogenic global warming (Mohamed et al., 2024; Oliver, Donat, et al., 2018; Wang & Zhou, 2024). Beyond long-term warming trends, recent studies have highlighted the importance of large-scale climate variability in enhancing MHW characteristics globally through atmospheric and oceanic teleconnections (Holbrook et al., 2019; Wang & Zhou, 2024). For example, El Niño-Southern Oscillation (ENSO) influences MHWs over the tropical Pacific and Indian Oceans (Hamdeno et al., 2024; Liu et al., 2022; Mohamed et al., 2022; Oliver, Perkins-Kirkpatrick, et al., 2018; Vivekanandan et al., 2008). During El Niño phases, the significant increases in sea surface temperature (SST) lead to prolonged MHWs in the eastern tropical Pacific (Podesta & Glynn, 2001). Similarly, the North Atlantic Oscillation (NAO) primarily affects the tropical and North Atlantic MHWs (Gröger et al., 2024; Holbrook et al., 2019; Mohamed et al., 2023; Scannell et al., 2016). Although current forecast systems show skills in predicting the occurrence of MHWs in the El Niño region (de Boisséson & Balmaseda, 2024), with prediction accuracy significantly improving during El Niño events (Jacox et al., 2022), the forecast accuracy remains poor in the Northeastern Atlantic Ocean influenced by NAO, particularly in the North Sea (de Boisséson & Balmaseda, 2024; McAdam et al., 2023). This limitation warrants further investigations into the mechanisms driving MHWs at the relevant spatio-temporal

505152

65

The North Sea is a shelf sea located on the passive continental margin of northwest Europe, connecting the Baltic Sea to the Atlantic. The SST and MHW patterns in this region are influenced by large-scale North Atlantic climate variabilities, including NAO, the Atlantic Multidecadal Variability (AMV) and East Atlantic Pattern (EAP) (Mohamed et al., 2023; Scannell et al., 2016; van der Molen & Pätsch, 2022). Distinct seasonal differences of MHW mechanisms and climate variability responses in the North Sea have been documented (Gröger et al., 2024; Mohamed et al., 2023). Summer and winter exhibit contrasting patterns: during summer, MHWs are more frequent with higher intensity but shorter duration, while winter shows fewer events with lower intensity but longer duration. The NAO and EAP represent the two dominant modes of atmospheric circulation variability over the Euro-Atlantic region (Thornton et al., 2023), while AMV refers to large-scale multidecadal fluctuations in Atlantic SST (Kerr, 2000). These climate variabilities regulate westerly winds in the North Atlantic, affecting warm Atlantic inflow into the southern North Sea (van der Molen & Pätsch, 2022). Existing studies show that MHW frequency in the southern North Sea increases during the positive phase of AMV or EAP, while positive NAO primarily intensifies winter MHW occurrence (Mohamed et al., 2023; Scannell et al., 2016). The southern North Sea, characterized by shallow depths (Fig. 1a), experiences low-frequency, high-intensity, and long-duration MHWs. In contrast, the northern North Sea, which features larger water depth, exhibits more frequent and intense, but shorter MHWs (Chen & Staneva, 2024). However, the underlying dynamic mechanisms remain poorly understood (Mohamed et al., 2023).

Moreover, historical studies have indicated that climate variabilities in the Atlantic Ocean are interconnected (Börgel et al., 2020; Delworth & Zeng, 2016; Delworth et al., 2017; Sun et al., 2015). For example, the AMV alters the zonal position of NAO centers of action (Börgel et al., 2020), while the NAO-Atlantic Meridional Overturning Circulation interaction shapes the AMV (Delworth & Zeng, 2016; Delworth et al., 2017; Sun et al., 2015). These Atlantic climate variations also connect with Pacific Ocean patterns (Oshika et al., 2015). The AMV modulates the impact of Arctic Oscillation on ENSO and tropical climate variability (Chen et al., 2025; Xue et al., 2025). Conversely, ENSO-induced dipolar convection anomalies exert an influence on NAO and EAP (Brönnimann, 2007; Hou et al., 2023; Jiménez-Esteve & Domeisen, 2018; Scaife et al., 2024; Wicker et al., 2024). Existing literature (Gröger et al., 2024; Hamdeno et al., 2024; Liu et al., 2022; Mohamed et al., 2023) has focused primarily on the effects of individual climate variabilities on MHWs, leaving their synergistic influence largely unexplored.

In this study, we aim to fill the knowledge gap in understanding synergistic impacts of climate variabilities on MHWs in shelf seas. We focus on the greater North Sea region, which is projected to warm as fast as global levels (Hobday & Pecl, 2014) but exhibits large climate variability caused by interactions between the Arctic and subtropical zones (Quante & Colijn, 2016). This makes it an ideal regional example for understanding how climate variabilities synergistically influence MHWs in complex shelf sea settings.

**Figure 1.** (a) The research domain of the North Sea. The bathymetry (m) is shown on a logarithmic scale. (b) Power spectrum analysis of marine heatwave cumulative intensity (MHWCI) anomalies in the North Sea derived from OSTIA. Orange dots indicate periods exceeding the 90% confidence level (black dashed line), while blue dots represent periods below this threshold.

### 2 Data and Methods

#### 2.1 Data

Our analysis of MHWs in the North Sea (Fig. 2a) is based on high-resolution SST data  $(0.05^{\circ} \times 0.05^{\circ}$ , daily) from the Copernicus Marine Environment Monitoring Service (CMEMS). This dataset integrates multiple satellite measurements and in-situ observations by the Operational Sea Surface Temperature and Sea Ice Analysis (OSTIA (Good et al., 2020), covering the period 1982-2021. Another observation-based SST data used for validation is from the FerryBox (Macovei et al., 2021), which was developed by Helmholtz-Zentrum Hereon (Fig. A1).

Atmospheric conditions were characterized using the ERA5 reanalysis dataset (Hersbach et al., 2020) from the European Centre for Medium-Range Weather Forecasts (ECMWF), which provides daily variables at  $0.25^{\circ} \times 0.25^{\circ}$ 

resolution from 1970 to present. These variables include 10-m wind components, 500-hPa geopotential height, precipitation, and heat flux. For oceanic variables, including potential temperature and current velocity, we used monthly data from the ECMWF Ocean Reanalysis System 5 (ORAS5).

#### 2.2 Cumulative intensity of MHW

MHWs, which are categorized as a thermal event when their associated temperature is larger than the 90th percentile threshold for at least 5 days (Hobday et al., 2016), are detected by using the MATLAB Marine Heatwaves (M\_MHW) toolbox (Zhao & Marin, 2019). The long-term warming trend was subtracted to better estimate the effects of climate variability (Liu et al., 2022). The MHW metrics include frequency, duration, mean intensity, and cumulative intensity. The potential impact of individual events on marine ecosystems is best represented by their cumulative intensity, combining the effect of duration and average intensity (Marin et al., 2021). In this study, the cumulative intensity was summed to derive the monthly cumulative intensity (MHWCI), which represents the intensity over the duration of MHW events occurring within one month (Gröger et al., 2024; Mohamed et al., 2023).

#### 2.3 K-means Clustering Analysis

Distinct regional patterns of MHWCI variability in the North Sea were identified using a correlation-based K-means clustering approach (Jain, 2010; Lloyd, 1982). Unlike the conventional K-means that minimizes the Euclidean distance, the correlation-based version groups grid points according to the similarity in the shape of their normalized feature vectors, thereby emphasizing pattern similarity rather than absolute amplitude. Such correlation-based K-means clustering has been successfully applied in gene expression data analyses (Loganantharaj et al., 2006) and brain connectivity dynamics (Allen et al., 2014), to identify regions with coherent temporal or spatial variability patterns.

Given the pronounced seasonal variability of MHW in this region, this approach classifies areas according to their seasonal patterns of MHWCI interannual variability, reflecting different regional responses to large-scale climate forcing. For each grid point, MHWCI anomalies were computed for each season by subtracting the long-term seasonal climatology. The four seasons were defined as winter (December of the previous year and January–February of the current year, named as DJF), spring (from March to May, named as MAM), summer (from June to August, named as JJA), and autumn (from September to November, named as SON). The interannual variability during each season was quantified as the standard deviation of anomalies across the 40 years, yielding a four-component feature vector per grid point, with each component representing DJF, MAM, JJA, and SON variability. To reduce the influence of extreme values and ensure equal weighting among the four seasonal components, a logarithmic transformation followed by standardization was applied to the feature vectors. The dissimilarity between two grid points (Eq. 1) was defined as the correlation distance between their feature vectors:

$$D(x_i, x_i) = 1 - r(x_i, x_i), \tag{1}$$

where  $r(x_i, x_j)$  is the Pearson correlation coefficient between the seasonal variability patterns of MHWCI, emphasizing the similarity in their seasonal variability structures. The total within-cluster dissimilarity was computed as the sum of squared correlation distances (WCSS), which decreases as the number of clusters K increases. The number of clusters K was determined using the Elbow Method, where the reduction in WCSS begins to level off (Syakur et al., 2018). In this study, K=2 was selected as the optimal number of clusters (Figure B1). Further validation of clustering robustness is provided in Appendix B.

Based on the assessment, grid points were partitioned into two clusters by grouping them with similar seasonal patterns. Each cluster is characterized by its dominant season, defined as the season with the highest mean standardized variability, and the relative contributions of all four seasons. This approach identifies regions where MHWCI variability exhibits specific seasonal dependence, thereby revealing distinct spatial patterns that may be linked to different large-scale climate variations.

#### 2.4 Heat budget analysis

The physical processes driving MHWCI variations in the subregions identified by the K-means clustering were quantitatively examined through an upper-ocean heat budget analysis. After linearizing the variables, the perturbation equation of upper ocean heat anomalies (Eq. 2) can be described as:

154 
$$\frac{\partial \mathbf{T}'_{m}}{\partial \mathbf{t}} = \frac{Q'_{net}}{\rho C_{p} \mathbf{h}_{m}} - \left( \vec{\mathbf{u}}_{g} \cdot \vec{\nabla} \mathbf{T}'_{m} + \vec{u}'_{g} \cdot \vec{\nabla} \mathbf{T}_{m} \right) - \left( \vec{\mathbf{u}}_{e} \cdot \vec{\nabla} \mathbf{T}'_{m} + \vec{u}'_{e} \cdot \vec{\nabla} \mathbf{T}_{m} \right) - \left( \overline{\mathbf{w}}_{e} \frac{(\mathbf{T}_{m} - \mathbf{T}_{d})'}{\mathbf{h}_{m}} - \mathbf{w}'_{e} \frac{\overline{(\mathbf{T}_{m} - \mathbf{T}_{d})}}{\mathbf{h}_{m}} \right) + 155 \quad RES,$$
(2)

where the overbar and prime denote monthly mean and anomaly, respectively. The left-hand side represents the sea surface temperature anomaly tendency. We calculated this tendency as the temperature difference between February and November for winter and between August and May for summer, respectively. The right-hand side comprises four main terms, namely the net heat flux, geostrophic heat advection, Ekman heat advection, and entrainment heat flux, respectively. The last term denotes the residual term (*RES*), which includes nonlinear heat advection and diffusion, etc. These terms are averaged over the respective seasonal periods.  $Q'_{net}$  refers to the sea surface net heat flux anomaly, comprising the net shortwave and longwave radiative components, as well as the net latent and sensible heat fluxes. Positive values of  $Q'_{net}$  indicate ocean gaining heat from the atmosphere.  $\rho$  is the seawater density (1025 kg m<sup>-3</sup>),  $C_p$  is the specific heat capacity (3890 J kg<sup>-1</sup> K<sup>-1</sup>), and  $h_m$  is the mixed layer depth (MLD), defined as the depth where temperature decreases by 0.5°C from the surface value (Monterey & Levitus, 1997). The horizontal circulation in the surface layer can be divided into the Ekman current component  $\vec{u}_e$  and the geostrophic current component  $\vec{u}_g$ .  $w_e$  is the Ekman pumping (entrainment) velocity, with positive values indicating Ekman upwelling, pumping or divergence and negative values for Ekman downwelling or convergence, respectively.  $T_d$  is the temperature below the base of the mixed layer.

#### 2.5 Horizontal wave activity flux

Large-scale atmospheric teleconnections influence the formation and persistence of MHWs by modulating atmospheric circulation and surface heat fluxes. The Pacific-Atlantic teleconnection is closely linked to Rossby wave propagation (Hou et al., 2023). The propagation of these teleconnection signals was diagnosed using the wave activity flux (WAF, Eq. 3), which quantifies the direction and intensity of stationary Rossby wave energy propagation (Takaya & Nakamura, 2001). Hence, the horizontal WAF (unit: m² s⁻²) was calculated as follows:

$$WAF = \frac{p\cos\varphi}{2|U|} \begin{cases} \frac{U}{a^2\cos^2\varphi} \left[ \left( \frac{\partial\psi'}{\partial\lambda} \right)^2 - \psi' \frac{\partial^2\psi'}{\partial\lambda^2} \right] + \frac{V}{a^2\cos\varphi} \left( \frac{\partial\psi'}{\partial\lambda} \frac{\partial\psi'}{\partial\varphi} - \psi' \frac{\partial^2\psi'}{\partial\lambda\partial\varphi} \right) \\ \frac{U}{a^2\cos\varphi} \left( \frac{\partial\psi'}{\partial\lambda} \frac{\partial\psi'}{\partial\varphi} - \psi' \frac{\partial^2\psi'}{\partial\lambda\partial\varphi} \right) + \frac{V}{a^2} \left[ \left( \frac{\partial\psi'}{\partial\varphi} \right)^2 - \psi' \frac{\partial^2\psi'}{\partial\varphi^2} \right] \end{cases} , \tag{3}$$

where p is the pressure normalized to  $1000 \, \text{hPa}$ ,  $\varphi$  is the latitude,  $\lambda$  is the longitude, a is the radius of the Earth,  $\psi(=\varphi/f)$  is the geostrophic stream function,  $\varphi$  (m) is the geopotential height, f is the Coriolis parameter and  $\psi'$  is the perturbed stream function. |U|, U and V represent the averaged wind speed, zonal, and meridional wind velocity, respectively.

#### 3 Results

### 3.1 Seasonal-to-decadal variability of MHWs

Power spectrum analysis of the detrended monthly MHWCI reveals two distinct temporal patterns, namely a seasonal variation and an interannual variation, both significant at the 90% confidence level (Fig. 1b). Based on the seasonal patterns of MHWCI interannual variability, the K-means clustering analysis identified two distinct subregions (named as Cluster 1 and 2, respectively) in the North Sea (Fig. 2a). Cluster 1 mainly covers the central and southern parts of the North Sea, while Cluster 2 includes the deeper northern part. Seasonal differences between the two clusters were quantified by calculating the relative contribution of each season, expressed as the percentage of its spatially averaged interannual variability intensity (VI) relative to all four seasons within each cluster. Positive values of VI indicate seasons that contribute more strongly to the total variability of that cluster, while negative values indicate weaker contributions.

Cluster 1 shows the highest variability in winter (VI = 0.66) and autumn (VI = 0.41), but lower variability in summer (VI = -0.28) and spring (VI = 0.07) (Fig. 2b). In contrast, Cluster 2 exhibits strong variability in summer (VI = 0.57) and moderate variability in spring (VI = 0.24), while autumn (VI = -0.08) and winter (VI = -0.30) are characterized by low variability. These results highlight spatially distinct seasonal responses showing that interannual variability in the central-southern shallow regions (Cluster 1) predominantly occurs in winter, whereas in the northern deeper areas (Cluster 2) interannual variability mainly occurs in summer.

The relationship between regional differences and large-scale climate variabilities was examined using the time series of domain-averaged MHWCI extracted for each cluster. Consistent with the identified seasonal dominance, the winter series was analyzed for Cluster 1 and the summer series for Cluster 2.

In Cluster 1 (mainly central and southern North Sea), negative NAO phases in late autumn (October, blue shading in Fig. 2c) are frequently followed by a positive winter EAP, which is significantly correlated with enhanced winter MHWCI (r=0.64, 95% confidence interval). For Cluster 2 (mainly northern North Sea), the summer-averaged MHWCI demonstrates distinct responses depending on the phase of the AMV. During the negative AMV phase (1982-1994, blue shading in Fig. 2d), MHWCI exhibits relatively weak intensity. In contrast, during the positive AMV phase (1994-2013, red shading in Fig. 2d), MHWCI shows notable enhancement, with its interannual variability strongly correlated with both ENSO and the Interdecadal Pacific Oscillation (IPO) (r=0.69 and 0.70, respectively). However, this teleconnection substantially weakens after 2013, with correlations becoming statistically insignificant (p>0.1), indicating a substantial reduction in Pacific influence on North Sea MHWs.

 These results indicate that the dominant climate drivers of MHWCI variability differ across regions and timescales: Cluster 1 is primarily shaped by late autumn NAO in combination with winter EAP, while Cluster 2 is modulated by summer Pacific telecommunications that vary with AMV phase. This spatial and temporal diversity suggests that MHWCI in the southern and northern parts of the North Sea is driven by distinct atmospheric circulation patterns and associated oceanic processes.

**Figure 2.** K-means clustering of marine heatwave cumulative intensity (MHWCI) in the North Sea and associated climate variability. (a) Spatial distribution of two clusters based on the seasonal patterns of interannual MHWCI variability. (b) Seasonal variability of each cluster. Positive values indicate seasons with stronger contributions to interannual variability, while negative values indicate weaker contributions. (c) Normalized time series of domain-averaged MHWCI for Cluster 1 during winter (black) and the East Atlantic Pattern (EAP, red); blue shading denotes negative NAO phases in October of the preceding year. (d) Normalized time series of domain-averaged MHWCI for Cluster 2 during summer (black) and Interdecadal Pacific Oscillation (IPO) index (red); blue and red shading indicate negative and positive phases of Atlantic Multidecadal Variability (AMV), respectively. The blue bar from 2013-2015 indicates the Atlantic "cold blob" event.

Composite analysis of MHW frequency, intensity, and duration anomalies during positive MHWCI periods highlights the underlying mechanisms (Fig. 3). In Cluster 1, the winter MHWCI increase is driven jointly by higher frequency, intensity, and duration (Fig. 3a-c). In Cluster 2, summer MHWCI enhancement is characterized by increased frequency and duration but decreased intensity (Fig. 3d-f), indicating that summer conditions promote persistent but less intense MHWs in the northern North Sea.

Figure 3. Composite anomalies of MHW characteristics during positive phases of the cluster-specific time series in winter (a-c) and summer (d-f). (a, d) MHW number, (b, e) MHW intensity (°C), and (c, f) MHW duration (days).

### 3.2 Cluster 1: Winter Mechanisms in the Central and Southern North Sea

The influence of climate variability on winter MHWCI in Cluster 1 through atmospheric processes was examined by compositing atmospheric variables during periods of enhanced MHWCI (positive phase of black line in Fig. 2c), including 500-hPa geopotential height, wind stress, WAF, and heat flux anomalies.

In October, the geopotential height anomalies exhibit pronounced negative anomalies over the subpolar Atlantic, with positive anomalies observed over the subtropical and polar regions (Fig. 4a). This atmospheric configuration corresponds to a negative NAO phase, characterized by a weakened Azores High and intensified Icelandic Low. The resulting cyclonic wind anomalies decrease SST over the central North Atlantic (Fig. 5a).

In November, the system enters a transition phase. SST cooling extends northward (Fig. 5b) under the influence of cyclonic circulation anomalies. Meanwhile, negative geopotential height anomalies over the subpolar and

polar Atlantic intensify, while positive anomalies extend northeastward. An eastward-propagating Rossby wave train develops, with WAF originating from the North American coast and propagating toward western Europe (Fig. 4b).

255256257

258259

260 261

262

254

By December, the wave train includes both eastward and westward-propagating components across the North Atlantic. This WAF configuration generates strong convergence over the subpolar region, significantly reinforcing the negative geopotential height anomalies there. Consequently, the atmospheric pattern transitions to a positive EAP phase. The geopotential height field displays a distinct tripole structure, reflecting intensified Azores High and Icelandic Low systems, accompanied by a strengthened high-pressure system over western Europe (Fig. 4c). The associated cyclonic winds maintain reduced SST in the subpolar region (Fig. 5c). This configuration persists into January (Fig. 4d), sustaining southwesterly wind anomalies over the northeastern North Atlantic (Fig. 5d).

263264

265266267

**Figure 4.** Composite anomalies of 500-hPa geopotential height (shading,  $m^2/s^2$ ) and horizontal wave activity flux (WAF, vector,  $m^2/s^2$ ) from October of the preceding year to January of the following year, based on winters with positive values in the cluster-specific time series. The anomalous WAF flux is shown only when its magnitude is larger than  $0.1 \text{ m}^2/s^2$ .

269270

Figure 5. Composite anomalies of sea surface temperature (shading, °C) and wind stress (vectors, N/m²) from October of the preceding year to January of the following year, based on winters with positive values in the cluster-specific time series.

These southwesterly winds enhance sensible heat loss from the ocean to the atmosphere, which serves as the primary contributor to negative net heat flux anomalies (blue bar in Fig. 6). By contrast, these southwesterly wind anomalies modify the ocean current convergence over the southern North Sea (Fig. 7b), enhancing SST. To further investigate the ocean response, circulation anomalies were composited during periods of enhanced MHWCI. The strengthened subtropical westerlies since late autumn enhance the transport of warm North Atlantic waters (Fig. 7a), increasing northward heat transport into the North Sea through the English Channel (Mohamed et al., 2023; van der Molen & Pätsch, 2022), and raising SST in the southern North Sea. After entering the North Sea, anomalous southwesterly winds drive this warm water northward, enhancing both meridional and zonal overturning circulations (MOC and ZOC) in the central North Sea (Fig. 7d, e). These strengthen oceanic downwelling and heat content (Fig. 7c), further offsetting the cooling from reduced net heat flux, illustrating the combined atmospheric and oceanic mechanisms driving winter MHWCI in Cluster 1. Additionally, the warm water inflow reduces cold outflow from the Baltic Sea.

**Figure 6.** Composite anomalies of net heat flux and its components (W/m<sup>2</sup>) in the North Sea during winters (blue bars) and summers (red bars) with positive values in the cluster-specific time series. Components include shortwave ( $Q_{sw}$ ) and longwave ( $Q_{lw}$ ) radiation, as well as the net latent ( $Q_{la}$ ) and sensible ( $Q_{sen}$ ) heat flux anomalies. Positive values indicate ocean heat gain from the atmosphere, while negative values indicate heat loss.

**Figure 7.** Composite anomalies of oceanic variables during positive phases of the cluster-specific time series in winter (a-e) and summer (f-j). (a), (f) Geostrophic oceanic current (vectors, m/s), (b), (g) convergence and divergence of advection (s<sup>-1</sup>), (c), (h) ocean heat content (J), (d), (i) zonal overturning circulations (ZOC, Sv), (e), (j) meridional overturning circulations (MOC, Sv). The black arrow-headed lines represent the direction of flow.

The upper-ocean heat budget for Cluster 1 was examined to evaluate the relative contributions of different mechanisms. Result shows that winter warming in the southern-central North Sea is primarily driven by geostrophic heat advection (Fig. 8), consistent with the enhanced Atlantic inflow through the English Channel (Fig. 7a). These results indicate that winter MHW frequency, intensity and duration in Cluster 1 are largely influenced by the North Atlantic climate variabilities through both atmospheric and oceanic pathways via enhanced westerly winds and strengthened Atlantic inflow.

Figure 8. Quantitative contribution (°C/season) of atmospheric and oceanic processes to the upper ocean temperature anomaly tendency for two clusters identified by the K-means algorithm. The atmospheric and oceanic variables are shown as composite patterns during positive phases of the cluster-specific time series in winter (blue) and summer (red). The temperature tendency (dT'/dt) represents the difference in the sea surface temperature anomaly between February and November for winter and between August and May for summer, respectively. Contributions from the net heat flux anomaly  $(Q_{net})$ , geostrophic heat advection  $(Adv_{Geo})$ , Ekman heat advection  $(Adv_{Ek})$ , entrainment heat flux (EHF), and residual term (RES), averaged over the same periods, are also shown.

#### 3.3 Cluster 2: Summer Mechanisms in the Northern North Sea

We identified the synergetic impact of AMV and Pacific telecommunications on the summer MHWCI in the North Sea. Comparison of atmospheric variable regression patterns onto the IPO index reveals stronger atmospheric responses in the North Pacific Ocean during strong positive AMV phases (1994-2012). The possible underlying mechanism can be explained through a sequence of atmospheric interactions. First, positive AMV induces anomalous ascent over the North Atlantic and anomalous descent over the North Pacific, which forms a dipole pattern of geopotential height with positive anomalies over the subpolar Pacific and negative anomalies over the subtropical Pacific (Fig. 9a). These interactions induce easterly wind anomalies between these regions (Lin et al., 2023). As a result, an anomalous cyclonic wind pattern emerges over the subtropical Pacific, strengthening ENSO-associated westerly winds and enhancing SST in the tropical Pacific Ocean (Hou et al., 2023). The associated warming along the North American west coast is consistent with the positive IPO phase (Fig. 10a).

The enhanced tropical and subtropical Pacific responses are transmitted across ocean basins through Rossby wave energy propagation. The WAF vectors clearly indicate the propagation of Rossby wave energy from the Pacific toward the North Atlantic, particularly under the positive AMV phase (Fig. 9a). This intensified wave propagation facilitates more efficient transmission of atmospheric perturbations from the Pacific, across North America, and into the Atlantic region. As part of this teleconnection pattern, northeasterly wind anomalies develop over the Caribbean Sea, reducing precipitation and SST in this region (Fig. 10b). The suppressed convection modifies the regional Rossby wave source, contributing to the maintenance of the anticyclonic pattern over the subtropical western North Atlantic. The associated southeasterly wind anomalies over the northwestern North Sea suppress Atlantic moisture transport, reducing cloud cover the over the northern North Sea (Fig. 10c), which increases downward shortwave radiation (red bars in Fig. 5), further contributing to enhanced downward net surface heat flux.

Heat budget analysis further confirms that in summer the MHW response in Cluster 2 is predominated by the surface heat flux and the RES term, whereas horizontal and vertical advection are comparatively weak (Fig. 8). During summer, the MLD is extremely shallow in Cluster 2 (13.2 m on average, decreasing to 12.5 m during positive phases), in contrast to much deeper winter MLD in Cluster 1 (88.9 m on average, and 84.9 m for positive phases). This shallow, strongly stratified surface layer enables short-lived radiative anomalies to trigger and prolong MHWs, while limited oceanic heat advection constrains subsurface heat storage, consistent with the absence of notable intensification in MHW intensity.

The easterly wind anomalies counter the prevailing climatological southwesterly winds. The resulting wind weakening generates cyclonic circulation anomalies in the northern North Sea (Fig. 7f), causing anomalous convergence of horizontal flows (Fig. 7g). These anomalous eastward inflows entering the North Sea produce positive ZOC anomalies (Fig. 7i), increasing ocean heat content (Fig. 7h). Although these circulation changes contribute locally to heat accumulation, their contribution to the overall heat budget is minor, indicating that geostrophic advection exerts only a limited influence on MHWCI variability in the Cluster 2 region (Fig. 8).

After 2013, the cold anomaly developed in the Atlantic Ocean (Mooney, 2015), coinciding with a weakening AMV signal (Frajka-Williams et al., 2017). During this period, the tropical and subtropical Pacific exhibit markedly diminished atmospheric responses (Fig. 10d-f). Tropical Pacific SST anomalies weaken substantially compared to 1994-2012, accompanied by the reductions in precipitation. Moreover, the geopotential height anomalies in the North Pacific weaken significantly (Fig. 9b), and the dipole circulation pattern between subpolar and subtropical Pacific substantially weakens. WAF analysis reveals a marked disruption of Rossby wave energy propagation from the Pacific to the Atlantic (Fig. 9b).

This Pacific-Atlantic teleconnection weakens even further during the negative AMV phase. The geopotential height dipole in the Pacific is much less pronounced (Fig. 9c), further diminishing the dipole circulation and reducing zonal wind anomalies between these regions. The SST anomaly distribution remains inconsistent with the IPO spatial pattern, with the North American coastal warming signature nearly absent (Fig. 10g). Correspondingly, the westerly

368 369

370

371 372

373

374 375

wind anomalies over the tropical Pacific are substantially more suppressed. This also suppresses the Rossby wave source, leading to a marked reduction of eastward Rossby wave energy flux into the Atlantic region (Fig. 9c). The diminished atmospheric response produces minimal impacts on North Atlantic circulation and limited modification of North Sea (Fig. 10 h-i).

**Figure 9.** Summer Composite patterns of 500-hPa geopotential height anomalies (shading,  $m^2/s^2$ ) and horizontal WAF (vector,  $m^2/s^2$ ) during (a) positive AMV phase (1994-2012), (b) positive AMV phase (2013-2021), and (c) negative AMV phase. All fields are regressed onto the IPO index. Only significant anomalies at the 90% confidence level are shown.

**Figure 10.** Summer atmospheric patterns regressed to the IPO index during different AMV phases. Composite anomalies during positive AMV phase (1994-2012) (a-c), positive AMV phase (2013-2021) (d-f), and negative AMV phase (g-i): (a), (d), (g) wind stress (vectors, N/m²) and sea surface temperature (SST, shading, °C), (b), (e), (h) precipitation (cm), and (c), (f), (i) cloud cover. Only significant anomalies at the 90% confidence level are shown.

### 4 Discussion

#### 4.1 Contrasting climate influences in MHWs across the North Sea

Our analysis of observed SST data demonstrates how multiple climate variabilities interact to influence MHWs at a regional scale through seasonally distinct mechanisms. Different from previous studies that mainly applied clustering analysis to group heatwaves based on common characteristics (Artana et al., 2024; Chauhan et al., 2023), location (Hansen, 2024), and underlying dynamic drivers (Vogt et al., 2022), this study employed a correlation-based k-means clustering approach to further explore their spatial and temporal coherence. Our analysis identified two distinct regions in the North Sea, a southern shallow region and a northern deep region, each dominated by different climate variations and seasonal dynamics. Similar north-south spatial contrasts in MHW-induced stratification associated with water depth have also been reported in the North Sea (Chen et al., 2022).

In the central-southern shallow North Sea (Cluster 1), enhanced MHWCI is closely linked to positive winter EAP conditions, typically following a negative late-autumn NAO. Previous studies have shown that MHW frequency in the southern North Sea increases during positive phases of either NAO or EAP (Mohamed et al., 2023). Similar NAO-related influences have also been reported in the adjacent Baltic Sea (Gröger et al., 2024). Our results refine this understanding by identifying the positive winter EAP as the primary driver, enhancing not only the frequency but also intensity and duration.

Recent research also indicates that AMO plays a more important role than the NAO in influencing the frequency of summer MHWs in the southern North Sea (Mohamed et al., 2023). However, its role in the northern North Sea has received little attention. Our results show that in Cluster 2, representing the deeper northern North Sea, the enhanced MHW frequency and duration in summer is linked to the synergistic influence of the AMV and Pacific climate variabilities. These teleconnections propagated from the Pacific to the North Sea through Rossby wave energy propagation, which is consistent with studies emphasizing the role of Pacific teleconnections in shaping the North Atlantic atmospheric pattern (Hou et al., 2023).

To exclude the influence of long-term warming, the SST data were linearly detrended before computing the MHW metrics. Even after detrending, Cluster 2 exhibits increasing frequency and duration accompanied by a slight decrease in intensity during positive MHWCI periods. This result is consistent with basin-wide observations from 1993 to 2022, which show a declining trend of MHW intensity but increasing frequency and duration (Chen & Staneva, 2024). Such change is attributed to large-scale atmospheric circulation changes and regional oceanic processes rather than global warming. The reduction in intensity has been proposed to be related to the weakening of the North Atlantic Jet Stream and more frequent atmospheric blocking events (Woollings et al., 2018), which promote stagnant atmospheric conditions and persistent but less intense warm anomalies over the North Sea, while freshwater-salt exchange, stratification changes, and Baltic Sea inflow also modulate regional SST responses (Mathis & Pohlmann, 2014). As an additional possible mechanism, the interannual-decadal variability linked to the AMV and Pacific teleconnections could modulate the summer North Atlantic atmospheric pattern, which then changes MHW intensity in the northern North Sea.

### 4.2 Global implications and future perspectives

Current MHW forecast systems exhibit highest skill in the El Niño region, the Caribbean, the wider tropics, the north-eastern extra-tropical Pacific, and southwest of the extra-tropical basins. However, the skill is much lower in the western Mediterranean and rather poor in the North Sea whatever the forecast range (de Boisséson & Balmaseda, 2024; Jacox et al., 2022). This is likely due to the impact of unresolved atmospheric variability, limited representation of teleconnections and climate modes (Ardilouze et al., 2017; Patterson et al., 2022). Since climate indices serve as important forecasting factors influencing these events (Jacox et al., 2022; Mi et al., 2025), incorporating synergistic climate interactions into forecasting frameworks could enhance predictive skill for MHWs.

 Similar teleconnection mechanisms have been documented in other shelf seas and marginal seas, including the Baltic Sea (Gröger et al., 2024) and South China Sea (Deng et al., 2022). For instance, the Indian Ocean Basin-Wide index has been identified as a key predictor of long-term MHWs occurrence in the South China Sea (Mi et al., 2025), while including ENSO-related variability in the tropical Pacific may extend the duration of skillful forecasts of atmospheric patterns over the North Atlantic (Shackelford et al., 2025). Our results reveal clear regional and seasonal contrasts in how large-scale climate variations influence MHW variability across the North Sea. These contrasts highlight that MHW predictability in shelf seas depends on the timing, location, and combined effects of multiple climate drivers rather than their individual strength alone. Incorporating these spatially and seasonally varying relationships into forecast systems would allow models to improve the ability to forecast when and where MHWs are most likely to occur.

### 5 Conclusions

By applying a correlation-based k-means clustering approach, this study revealed coherent spatial and temporal patterns of MHWCI variability in the North Sea and their connections to large-scale climate variations. The results show that multiple interacting climate variabilities jointly drive MHWCI variability on the northeastern Atlantic shelf, emphasizing the need to consider combined effects rather than individual forcing mechanisms. The relative importance and mechanisms of these large-scale drivers vary regionally and seasonally: the EAP exerts stronger control in the southern North Sea during winter, through two pathways, namely an atmospheric bridge via enhanced southwesterly winds and an oceanic pathway through strengthened Atlantic inflow (Fig. 11a).

Deep 150m

Cluster 2

450 451

457 458

461 462

459

463

464

In contrast, the AMV and Pacific-related teleconnections dominate in the northern region during summer. During positive AMV phases, the dipole geopotential height pattern along the North American west coast, reinforces IPO-like structures and strengthens Pacific-Atlantic teleconnections. These teleconnections, propagated from the tropical and subtropical Pacific to the North Atlantic through Rossby wave energy propagation, ultimately influence cloud cover over the North Sea and alter sea-air heat fluxes (Fig. 11b). When the AMV is in transitions to neutral or negative phases, the weakening of IPO-like patterns reduces Pacific-Atlantic coupling, thereby diminishing their influence on the region. Together, these regionally and seasonally distinct mechanisms provide a clear physical basis for how large-scale climate variability modulates MHWs in shelf seas.

These findings have practical implications for enhancing process-based prediction and for understanding how MHW characteristics may respond to future changes in large-scale climate variability. Future research should test whether similar interaction patterns occur in other shelf seas, evaluate the potential of climate indices to improve MHW prediction skill, and investigate how the underlying mechanisms may change under future climate conditions.

466 467

468

470

Figure 11. Schematic illustration of two distinct cluster patterns showing the synergistic impacts of climate variability on MHWCI in the Northwestern European shelf. (a) Winter pattern characterized by a combined negative NAO phase in late autumn and a positive EAP phase in winter. (b) Summer pattern characterized by simultaneous positive phases of AMV, PDO, and El Niño. Purple arrows indicate wind anomalies. Red and yellow circles denote high and lowpressure systems, respectively: AH refers to the Azores High, WEH to the West European High, and IL to the Icelandic Low. The red box over the North Sea highlights regions of enhanced MHWCI. Red upward and blue downward arrows represent increases and decreases in the associated variables. Blue cloud shapes denote cloud cover (CC), while white cloud shapes represent precipitation (Pre).  $Q_{sw}$  indicates the downward shortwave heat flux. Green arrows mark horizontal flow convergence or the Meridional Overturning Circulation (MOC).

## Appendices A: Validation of SST Data

→ Wind

In Sect. 2.1, The high-resolution SST data from OSTIA (Good et al., 2020) is validated by the FerryBox (Macovei et al., 2021), which was developed by Helmholtz-Zentrum Hereon. The FerryBox, installed on commercial vessels, records measurements every 20 seconds along fixed routes, providing data at approximately 100-meter spatial resolution. Our validation is focused on two specific routes: "Hafnia20160120" (0.2°W-8.7°E, 52.6-54.9°N) and

"Lysbris20170407" (0.2°W-12.8°E, 51.3-59.4°N). Figure A1 shows high consistency between these two datasets with root mean square error (RMSE) of 0.7 and 1.3 degrees for the year 2016 and 2017, respectively.

**Figure. A1.** Comparison of SST from the Operational Sea Surface Temperature and Sea Ice Analysis (OSTIA) dataset (blue line) with FerryBox in-situ measurements (red dots). Our validation focused on two specific routes: (a) "Hafnia20160120" (0.2°W-8.7°E, 52.6-54.9°N) and (b) "Lysbris20170407" (0.2°W-12.8°E, 51.3-59.4°N).

#### Appendices B: Clustering validation

The stability and interpretability of the correlation-based K-means clustering results were assessed through two complementary analyses: the elbow method and silhouette coefficient. The elbow curve method runs k-means clustering on the dataset for a range of K (from 1 to 10). For each of the K values, the total within-cluster sum of squared correlation distances (WCSS) (Lloyd, 1982) was computed. The optimal K corresponds to the point at which the decrease in distance becomes substantially less pronounced, indicating that further partitioning does not significantly improve intra-cluster cohesion. As shown in Figure B1, the elbow occurs at K = 2, suggesting that two clusters sufficiently capture the dominant spatial patterns of MHWCI variability in the region.

**Figure. B1**. Elbow method validation of K-means clustering of marine heatwave cumulative intensity (MHWCI) in the North Sea. The within-cluster sum of squared correlation distances (WCSS) is shown as a function of cluster numbers. The red circle indicates the optimal number of clusters determined by the Elbow method.

501 The mean silhouette coefficient (Rousseeuw, 1987) was then calculated for each K using the same correlation 502 distance metric. For a given observation i, the silhouette coefficient is defined as:  $S(i) = \frac{b(i)-a(i)}{\max\{a(i),b(i)\}}$ 503 (4) where a(i) is the average correlation distance between a point and all others in its own cluster, b(i) is the minimum 504 505 average distance to points in another cluster. The silhouette coefficient ranges from -1 to +1, with higher values 506 indicating better separation. It yielded a favorable score of 0.563 for k = 2, confirming the stability and physical 507 interpretability of the two-cluster partition. 508 509 Code availability 510 The code used for data analysis and visualization in this study is available at Zenodo with DOI: 511 https://doi.org/10.5281/zenodo.17358865 512 513 Data availability 514 All datasets are publicly available. Observational temperature datasets from OSTIA (Worsfold et al., 2024) were used 515 in this manuscript, available at: https://doi.org/10.48670/moi-00168. The Ferrybox data (Macovei et al., 2021) are available at https://doi.pangaea.de/10.1594/PANGAEA.930383. The ERA5 (Hersbach, 2023) and ORAS5 516 (Copernicus Climate Change Service, 2021) are provided by the Copernicus Climate Change Service, available at: 517 https://cds.climate.copernicus.eu/datasets/reanalysis-era5-single-levels-monthly-means?tab=overview 518 https://cds.climate.copernicus.eu/datasets/reanalysis-oras5?tab=overview. 519 520 521 **Author contribution** 522 Y.L. and W.Z. conceived the study together. Y.L. analyzed data and wrote the original draft under the supervision of 523 W.Z. Z.L., Z.F., Q.M., and W.Z. contributed to the revision of the text and helped to determine the final structure of 524 the article. 525 526 **Competing interests** 527 The authors declare that they have no conflict of interest. 528 Disclaimer 529 Publisher's note: Copernicus Publications remains neutral with regard to jurisdictional claims made in the text, 530 published maps, institutional affiliations, or any other geographical representation in this paper. While Copernicus Publications makes every effort to include appropriate place names, the final responsibility lies with the authors. 531 532 533 Acknowledgments 534 The authors would like to thank Dr. Yoanna Voynova and Dr. Vlad-Alexandru Macovei from Helmholtz-Zentrum 535 Hereon for providing information of the FerryBox installed on commercial vessels travelling in the North Sea. 536 537 Financial support 538 This study is a contribution to the BMBF MARE:N project 'Natural hazards and marine ECOsystem response - causal linkage and predictability (NECO)' (03F0950A: Y.L., W.Z.). It is also supported by the Helmholtz research 539 programme in PoF IV 'Changing Earth—Sustaining our Future'—'Topic 4: Coastal Transition Zones under Natural 540

- and Human Pressure' (W.Z.). This work is also supported by a grant from the Research Grants Council of the Hong
- Kong Special Administrative Region, China (Project Reference Number: AoE/P-601/23-N) and the National Natural
- Science Foundation of China (Grant numbers: 42230404 and U23A2023). Y.L. is supported by grants from the China
- Scholarship Council.

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

21 / 21