# Peer review of "Climate Variabilities Synergistically Influence Marine"

_EGUsphere, 2025_

## Author Comment (AC1)

The paper by Lin and coauthors deals with marine heatwaves in the North Sea. The contents and quality of the presentation need improvement. I have some doubts on the methodology and clarity of the diagnosed mechanisms. The Authors should consider the comments provided below.

**Response**: Thank you very much. We carefully considered each of your comments and suggestions, and we would like to respond to them in the following content.

I have also noticed this paper https://www.researchsquare.com/article/rs-6503093/v1 from the same authors, but with a different first author. Contents are similar but not identical. Could you please clarify the situation?

**Response**: Thank you for raising this point. The manuscript you referred to is a preprint that was previously submitted to another journal and subsequently posted online. That submission was unfortunately not accepted for publication. We would like to clarify that the first author of the preprint is also the first author of the present manuscript. The corresponding author's name appeared first in the header of that preprint, but in the manuscript the order of authors is consistent in both versions.

The present manuscript in Ocean Science represents a substantially improved and revised version of that earlier work. In preparing the current submission to Ocean Science, we have reanalyzed the results, clarified the scientific focus, revised the methodology, and reorganized the manuscript to better address the scope of Ocean Science. While the two manuscripts share a common scientific background, the current version contains significant modifications and improvements and is not under consideration elsewhere.

General comments

While there may be implication for prediction, mentioning forecasting in the abstract can be misleading. I suggest to clarify that is a diagnostic analysis, instead.

**Response**: Thank you for this helpful comment. We agree that our study is based on diagnostic and mechanistic analyses rather than forecasting. To avoid potential misunderstanding, we will revise the abstract by clarifying the diagnostic nature of the study and by removing wording that could be interpreted as implying direct forecasting.

"variabilities" in the plural form sounds strange. Suggest using singular or another noun, e.g. "modes" depending on what you want to convey.

**Response**: We thank the reviewer for pointing this out. We agree that the plural form "variabilities" is confusing in this context. We will revise the manuscript by replacing "climate variabilities" with "climate modes" to better reflect the physical meaning.

Nonlinear quantities are computed for both atmospheric and ocean reanalysis data; can you estimate the error made with this compared to using daily data?

**Response**: Thank you for this important comment. We acknowledge that computing nonlinear terms from monthly mean reanalysis fields may differ from first calculating these terms at daily resolution and then averaging, and that high-frequency variability can contribute to nonlinear processes. Our analysis focuses on seasonal to interannual variability, for which monthly-mean fields are commonly used. At these timescales, high-frequency synoptic fluctuations tend to partially cancel when averaged, and their net contribution to the low-frequency nonlinear terms is generally smaller than that of the resolved mean and low-frequency anomalies. As a result, our estimates should be interpreted as diagnostics of low-frequency, large-scale contributions, rather than an exact closure of the instantaneous nonlinear budget.

While using daily data could provide a more complete quantification of high-frequency nonlinear effects, we will prove in the revised version that this does not affect the main conclusions regarding the dominant processes at seasonal to interannual timescales.

I am not sure to understand your use of K-means in this work. In other works (e.g. Vogt et al. 2022 10.3389/fclim.2022.847995, Wong et al. 2024 https://agupubs.onlinelibrary.wiley.com/doi/10.1029/2023AV001059) the procedure is meant to identify groups of data points, while in your case two regions (Fig. 1) are identified.

**Response**: Thank you for this comment. We apologize that the explanation of the K-means analysis was not sufficiently clear in the original manuscript. In this study, K-means clustering is applied to the temporal evolution of MHW cumulative intensity (MHWCI) at each spatial grid point, rather than to individual MHW events or time steps. Each grid point is therefore represented by its MHWCI time series, and grid points that exhibit similar temporal variability are grouped by the clustering. As a result, the clusters naturally emerge as contiguous spatial regions (Fig. 1). We will revise the manuscript to better clarify this. Further validation of clustering robustness is provided in Appendix B. This analysis results in two well-separated and spatially coherent regions in the North Sea, broadly corresponding to the southern and northern subregions, indicating distinct temporal variability characteristics of MHWCI.

And from Fig. 3, I have the impression this just means seasonality. How can these two domains be related to MHW occurrence?

**Response**: Thank you for this comment. We agree that Fig. 3 highlights clear seasonal contrasts between the two domains. The K-means clustering is not applied to individual MHW events or occurrence number, but to the time series of MHWCI at each grid point. MHWCI integrates the occurrence, duration, and intensity of MHWs, and its seasonal evolution therefore provides a meaningful representation of MHW activity. By clustering grid points with similar seasonal and interannual variability of MHWCI, the analysis identifies regions that share common temporal characteristics of MHW activity. In this sense, the two domains are directly related to MHW occurrence through their distinct seasonal modulation of MHWCI, rather than being defined by climatological seasonality alone. We will add explanations in the revised version to clarify this.

From line 201 and following I understand MHWs may happen predominantly in one or the other region depending on the season, but more explanation is due.

**Response**: Thank you for this comment. We agree that additional explanation is needed. In the revised version, we will add an explanatory paragraph to explicitly link the seasonal diagnosis shown in Fig. 2b with the subsequent analysis strategy.

Various climate indices are shown; sources for the associated data is missing.

**Response**: We thank the reviewer for pointing this out. We will revise the manuscript to explicitly describe the data sources and calculation methods of all climate indices used in this study. A new subsection (Sect. 2.4) will be added to the Methods section, detailing the definitions, computational approaches, and data sources for the NAO, EAP, and AMV indices.

The discussion on the mechanisms is quite poor, as results for a much larger domain is presented, so local mechanisms are not discussed. More details should be provided on the stratification procedure;

**Response**: We thank the reviewer for the constructive comments. Regarding the discussion of mechanisms, the local processes were investigated through a detailed upper-ocean heat budget analysis conducted exclusively within the North Sea. However, we acknowledge that the current manuscript places stronger emphasis on large-scale circulation patterns and teleconnections, which may obscure the local nature of the diagnosed mechanisms. In the revision, we plan to reorganize the Results and Discussion sections to more explicitly highlight the heat budget analysis as the key representation of local physical mechanisms.

With respect to stratification, we understand the reviewer's comment as referring to vertical ocean stratification. Although stratification effects are implicitly included through the mixed layer depth (MLD) in the heat budget framework, this aspect was not sufficiently discussed in the current version. We plan to explicitly address the role of stratification and clarify how differences in MLD modulate the relative importance of surface heat fluxes and oceanic advection between the two clusters.

are you using fixed or dynamic thresholds? This should be consistent across indices, e.g. if they are all standardized already.

**Response**: We thank the reviewer for this helpful comment. We clarify that fixed thresholds are used throughout the analysis, but they are applied differently depending on the type of variable.

(1) Climate indices: The large-scale climate indices (NAO, EAP, and AMV) are obtained directly from publicly available datasets. These indices are anomaly-based indices derived from atmospheric or oceanic fields, and we use them in their original form without applying additional standardization. Positive and negative phases shown in Fig. 2 and Fig. 9 are therefore defined using a fixed threshold at zero, with values greater (less) than zero indicating positive (negative) phases.

(2) Cluster-specific MHWCI time series: The cluster-specific MHWCI time series are constructed in this study and are explicitly standardized to zero mean and unit variance. Periods of enhanced MHWCI are defined using a fixed threshold of +1 standard deviation, corresponding to values greater than 1 in the standardized time series. This threshold is applied consistently in all composite and regression analyses.

We will revise the Methods and Results sections to explicitly clarify these definitions and to avoid any ambiguity regarding the thresholds used.

*The level of discussion when presenting figures (which are often hard to read) is insufficient, and the various domains used (e.g., larger in Fig. 10) complicates comparisons.*

**Response**: We thank the reviewer for pointing out issues related to figure presentation and clarity. We agree that, in several cases, the discussion accompanying the figures does not sufficiently guide the reader through what is shown in the plots. In the revised manuscript, we plan to revise the Results section so that each figure is first described in a clear and descriptive manner (i.e., spatial patterns, temporal changes, and contrasts), before moving on to physical interpretations and discussion.

Regarding the use of different spatial domains (e.g., the larger domain in Fig. 10), we acknowledge that this may complicate direct comparison with other figures. To address this, we will explicitly highlight the key comparison regions by adding boxes and annotations in the figure and clarify in the captions and text. This will help guide the reader's attention and improve consistency across figures.

*I would suggest to first analyse non-MHW variables and understand how they change over time and due to teleconnections, and then focus on the influence of these indices on MHWs.*

**Response**: We appreciate this suggestion and agree that the current presentation may give the impression that the causal chain is not clearly structured. While our analyses of atmospheric and oceanic variables were conducted to diagnose the mechanisms underlying MHW variability, the manuscript currently introduces these results in direct connection with MHWs.

In the revision, we plan to improve the narrative flow by first more clearly describing the evolution of non-MHW variables (e.g., circulation, heat fluxes, mixed layer depth) and their modulation by large-scale climate modes, and then link these changes to the observed MHW responses. This restructuring will better reflect the physical causality and improve readability without altering the core analyses.

*I feel like the summary of Fig. 11 is actually not aiding interpretation, as new concepts and indices are added. This should be simplified.*

**Response**: We agree that the current summary figure (Fig. 11) introduces additional concepts and climate indices, which may complicate the interpretation. In the revised manuscript, we will simplify Fig. 11 so that it functions purely as a conceptual summary of the key mechanisms.

*Appendix A seems unnecessary, as this has likely been done by the producers in more detail.*

**Response**: We agree that the validation of the SST product has been extensively documented by the data producers. Following this recommendation, we will remove Appendix A from the manuscript. The validation will be mentioned in the Data section as supplementary information.

The work needs careful proofreading, as there are errors in the titles of most figures and references. Only some examples are given below.

**Response**: Thank you. We will carefully proofread the entire manuscript and correct typographical errors and inconsistencies in figure titles, references, and text throughout the paper.

Comments by line
l21 why speaking about "prediction"?

**Response**: We agree that the reference to "prediction" in the abstract is confusing. As the present study is based on diagnostic and mechanistic analyses rather than forecasting, we will revise the abstract to clarify its diagnostic nature and remove confusing wording.

l69 same comment as for the title

**Response**: We will revise the manuscript at line 69, replacing "climate variabilities" with "climate modes".

Fig. 1 typo in panel b title

**Response**: The typo in the title of Fig. 1b will be corrected from "MHWTI" to "MHWCI".

l88/184 as far as I understand, cumulative intensity is a per-event quantity, and events are discrete. How do you compute the Fourier spectrum then? Is it just SST?

**Response**: We thank the reviewer for this comment. We clarify that the Fourier spectrum is not computed from individual MHW events. While cumulative intensity is defined at the event level, we construct a continuous time series by aggregating the cumulative intensity of all MHW events occurring within each month. MHWCI is set to zero when no marine heatwaves occur during a given month. This results in a monthly MHWCI time series, which is then used for the spectral analysis shown in Fig. 1b. The analysis is therefore based on monthly MHWCI, not on SST directly. Clarification will be provided in the revision.

l102 The data record starts in 1940. Are you using 1982-2021 as for the SSTs? Please clarify

**Response**: The satellite-based OSTIA SST product used to identify MHW is available only from 1982 onwards. Although the atmospheric datasets (e.g. ERA5) extend back to 1940, our analysis is restricted to the common period 1982-2021, consistent with the SST record.

l104 A reference and some more details, both on ERA5 and ORAS5, should be given

**Response**: We thank the reviewer for this comment. We will revise the manuscript to include appropriate references and additional details for both ERA5 and ORAS5 in the Data section, including a brief description of the datasets and their temporal and spatial resolutions.

l113 MHWCI definition ain't very clear to me. What happens say if an event starts in Jan 27 and ends on Feb 10? Which month gets the CI? Please specify also the baseline period used.

**Response**: Thank you for this comment. We constructed monthly MHWCI by aggregating event cumulative intensity within each calendar month. Specifically, monthly MHWCI is obtained by summing the cumulative intensity of all events whose onset date falls in that month. Therefore, an event starting on 27 January and ending on 10 February is assigned to January in our monthly aggregation. We additionally tested the sensitivity of the k-means classification to the event-to-month assignment by assigning events to their termination month instead of their initiation month. The resulting spatial clustering remains very similar to the original classification. The main difference is a slight seasonal shift in the relative contributions within each cluster: in Cluster 1, which is primarily winter-dominated, the contribution from spring becomes somewhat larger, while in Cluster 2, which is primarily summer-dominated, the contribution from autumn increases. Importantly, however, the dominant seasons associated with MHW occurrence remain winter and summer, respectively. Therefore, this sensitivity does not affect the main physical interpretation or conclusions of the study.

Furthermore, the MHW threshold (90th percentile) is computed from the detrended SST time series using the baseline period 1982-2021. Explanations will be provided in the revision.

l154 state the source of this version of the heat tendency equation. Information on the number of levels in ORAS5 should be added, to clarify which is the accuracy of the MLD estimated as such eq3 I imagine these equation is also calculated with monthly mean data?

**Response**: Thank you for the comment. The mixed-layer heat tendency equation (Eq. 3) follows the standard formulation commonly used in previous studies of upper-ocean heat budgets (Liu et al., 2014; Tan et al., 2016). We will add the relevant references in the manuscript.

ORAS5 provides ocean variables on 75 vertical levels (level spacing increasing from 1 m at the surface to 200 m in the deep ocean), allowing a physically consistent estimation of mixed-layer depth and associated heat budget terms at seasonal to interannual timescales. All heat budget terms are computed using monthly mean data. We will clarify these points in the Methods section.

l189 so you apply K-means clustering over the whole year or individual seasons?

**Response:** The K-means clustering is not applied separately to individual seasons, nor directly to the raw monthly time series. Instead, it is performed using seasonal variability characteristics derived from the full monthly record. Specifically, for each grid point, we compute the interannual variability (standard deviation) of MHWCI anomalies separately for DJF, MAM, JJA, and SON (with DJF treated across calendar years). These four seasonal variability measures form a feature vector that is used as input to the K-means clustering. Therefore, the clustering is based on information from all seasons, while explicitly accounting for their distinct variability characteristics.

Fig. 2 typos in titles, and no units in Fig c and d; how are time series normalized?

**Response**: We thank the reviewer for pointing this out.

(1) We will correct the typos in the titles of Fig. 2c-d.

(2) We will clarify in the figure labels and captions that the time series shown in Fig. 2c and d are standardized MHWCI, which are therefore dimensionless.

(3) The time series are normalized using z-score standardization, defined as

$$z(t) = \frac{x(t) - \mu}{\sigma},$$

where $\mu$ and $\sigma$ denote the mean and standard deviation of the original time series, respectively. This procedure results in dimensionless time series with zero mean and unit variance.

l247 anomalies from what? Is this is stratified according to some indices? How?

**Response**: We thank the reviewer for this comment. The geopotential height anomalies shown in Fig. 4 are defined relative to the long-term monthly climatology. They are obtained using composite analysis during periods of enhanced MHWCI, defined as months when the standardized cluster-specific MHWCI time series exceeds +1 standard deviation. We will clarify the anomaly reference state and the composite criteria in the revised manuscript.

l266 with units m²/s², this is geopotential (not height)

**Response**: We thank the reviewer for pointing this out. We agree that the variable shown has units of m² s⁻² and therefore represents geopotential rather than geopotential height. We will revise the text and figure labels throughout the manuscript accordingly to ensure consistent terminology.

l269 why this value? Why not using some significance threshold?

**Response**: We thank the reviewer for this comment. The composite anomalies shown are tested for statistical significance at the 90% confidence level. For the wave activity flux vectors, in addition to significance testing, we apply a display threshold of 0.1 m² s⁻² in the figure to emphasize dynamically meaningful flux patterns. Similar display thresholds have been used in previous studies of atmospheric wave activity flux diagnostics (Hou et al., 2023). Explanation will be provided in the revision.

Fig 7 cannot be understood. Why are arrows colored? They are hard to see. What are the black arrows on the right side? Missing labels on axes

**Response**: We agree that Fig. 7 currently contains too many elements, which make it difficult to interpret. In the revised version, we will simplify the presentation of the vector fields by representing current speed through arrow length rather than color, while using a uniform color for all vectors. In addition, we will increase arrow size and spacing to further enhance readability. The black arrows on the right-hand side indicate the direction of the meridional overturning circulation. We agree that this was not sufficiently clear in the current version and will clarify this explicitly in both the caption and the main text. In addition, we will add missing axis labels and ensure that all plotted elements are clearly defined.

l277 is this some sort on average on some sub-domain?

**Response**: The contributions shown in Fig. 6 are computed as spatial averages over the Cluster 1 region during periods of enhanced MHWCI. We will clarify this in the revised manuscript.

l318 "telecommunications"?

**Response**: The term "telecommunications" was an error and will be corrected to "teleconnections" in the revised manuscript.

l340 "predominated"

**Response**: We thank the reviewer for pointing this out. The term "predominated" will be replaced with "dominated" in the revised manuscript.

l342 and otherwise? Some climatological maps should be presented.

**Response**: We understand that the concern refers to the description of mixed layer depth (MLD), where only regional mean values are reported, and more generally to the representation of stratification in the manuscript. In the revised manuscript, we will include climatological maps of MLD and discuss stratification together with MLD, linking this discussion to the heat budget analysis and to the contrasting summer and winter mechanisms identified in the two clusters.

Fig. 9 why do arrows look quite different? And what is the box in the maps?

**Response**: The differences in the WAF vectors among the three panels reflect physically distinct wave propagation patterns associated with different AMV phases and background climate states, as all fields are consistently regressed onto the IPO index. We will clarify this point in the figure caption. In addition, the rectangular box shown in Fig. 9 indicates the North Sea region, which is the focus of the present study. This will be stated in the caption.

l400 is it because some versions of the AMO index are influenced by trends?

**Response**: We thank the reviewer for this insightful question. We agree that some versions of the AMO index can be influenced by long-term trends. However, the AMO index used in Mohamed et al. (2023) is based on the NOAA index definition in which the North Atlantic SST time series is detrended and thus represents multidecadal internal variability rather than the long-term warming signal.

In our study, we detect marine heatwaves using detrended SST. Therefore, the relationships discussed here are unlikely to arise from shared long-term trends but instead reflect the influence of low-frequency Atlantic variability and associated atmosphere-ocean processes.

l425 why the West Med now?

**Response**: We agree that the reference to the western Mediterranean is not essential in this context and may appear abrupt. We will remove this part of the sentence in the revised manuscript to improve the focus and clarity of the discussion.

l662 incomplete citation, also l681, l700...

**Response**: We thank the reviewer for pointing this out. We will carefully check the entire reference list and correct all incomplete citations, including those at lines 662, 681, and 700.

Fig. B1 what are the units of the ordinate?

**Response**: The ordinate in Fig. B1 represents the within-cluster sum of squared correlation distances (WCSS), which is a dimensionless clustering metric. We will clarify this in the figure caption.

Reference used in the response

Hou, J., Fang, Z., & Geng, X. (2023). Recent Strengthening of the ENSO Influence on the Early Winter East Atlantic Pattern. *Atmosphere*, *14*(12), 1809. https://www.mdpi.com/2073-4433/14/12/1809

Liu, Q. Y., Wang, D., Wang, X., Shu, Y., Xie, Q., Chen, J., Liu, Q. Y., Wang, D., Wang, X., & Shu, Y. (2014). Thermal variations in the South China Sea associated with the eastern and central Pacific El Nino events and their mechanisms. *Journal of Geophysical Research Oceans*, *119*(12), 8955-8972.

Tan, W., Wang, X., Wang, W., Wang, C., & Zuo, J. (2016). Different Responses of Sea Surface Temperature in the South China Sea to Various El Niño Events during Boreal Autumn. *Journal of Climate*, *29*(3), 1127-1142.

---

## Author Comment (AC2)

General comments:
This paper presents an analysis of the factors that affect the occurrence of marine heatwaves in the North Sea. A correlation-based K-means clustering approach was used to divide the North Sea into two regions that have different variability in marine heatwave cumulative intensity, and detailed analysis carried out to determine the mechanisms that cause that variability. The paper is well written and presented, and the topic and results are of strong interest.

**Response**: Thank you very much. We carefully considered each of your comments and suggestions, and we would like to respond to them in the following content.

Specific comments:
Line 97 – should Worsfold et al. (2024; https://doi.org/10.3390/rs16183358) be referenced here?

**Response**: Thank you for noting this. We will add the reference.

Line 157 – a minor comment, but why include November and May in the calculation of tendency but not March or September? Shouldn't it be calculated either November - March and May – September if including the months surrounding the season, or December – February and June – August if not?

**Response**: Thank you for this comment. We agree that our description of the SST anomaly tendency window could be confusing. To ensure temporal consistency between the left-hand-side tendency and the seasonally averaged budget terms, we will revise the tendency definition to be computed within the core season: (Feb-Dec) for winter (DJF) and (Aug-Jun) for summer (JJA), normalized by the corresponding time interval. All right-hand-side terms are now averaged over the same DJF/JJA periods. The manuscript will be updated accordingly.

Line 184 – I struggle to relate the text here, which says two distinct temporal patterns, to Figure 1b, which does not seem to show a strong peak at ~6 month period. Perhaps a log scale on the y axis might help?

**Response**: Thank you for this important and constructive comment. We apologize for not clearly describing the preprocessing applied prior to the spectral analysis. The power spectrum shown in Figure 1b was computed from a 12-month running-mean MHWCI anomaly, which acts as a low-pass filter and was intended to emphasize interannual variability. As a result, seasonal variability is strongly dampened and is therefore not expected to appear as a pronounced spectral peak in this figure. To avoid confusion, we will revise this figure and corresponding texts in the manuscript.

Equation 3 – Is the left brace notation intended? The equivalent equation in the referenced paper uses square braces. Also, I'm not clear why the stationary version of the equation is used rather than the full one?

**Response**: Thank you for this comment. You are correct regarding the notation. We will revise Eq. (3) to use square brackets for consistency with the original presentation in Takaya and Nakamura

(2001). The wave activity flux formulation adopted here follows the stationary Rossby wave activity flux as defined by Takaya and Nakamura (2001), which is widely used to diagnose the propagation of quasi-stationary Rossby wave energy in seasonal-mean and large-scale circulation patterns. Since our analysis focuses on seasonally averaged atmospheric teleconnections rather than transient wave evolution, the stationary formulation is appropriate for the purpose of this study.

Line 185 – How was the 90% confidence interval calculated?

**Response**: The 90% confidence level in Fig. 1b was estimated using a chi-square significance test against a stochastic noise background. Specifically, we computed the power spectrum from the detrended monthly MHWCI anomalies (with Hanning smoothing), estimated the effective degrees of freedom, and then derived the 90% confidence threshold from the $\chi^2$ distribution. When the lag-1 autocorrelation is positive (and larger than lag-2), we adopt a red-noise (AR(1)) background spectrum following the standard formulation; otherwise a white-noise background is used. Spectral peaks exceeding this 90% threshold are marked as significant in the figure. We will clarify this procedure in the caption of Figure 1.

Line 211 – 2013 onwards seems a short period to calculate the correlation, even though it is statistically significant. Do any other similar length periods in the record have low correlation?

**Response**: Thank you for this comment. To address this concern, we calculated correlations using moving windows of the same length as the 2013-2022 period across the full record. The sliding-window analysis shows that correlations remain consistently high positive AMV phase (1994-2013), even when the same short window length is applied. In contrast, correlations are weak as those observed after 2013 (Fig. R1). This indicates that the 2013-2022 reduction is not a typical feature of short windows, but instead reflects a distinct weakening of the relationship. We will add this figure in the supplementary of revised manuscript.

[Figure]

**Figure R1**. Sliding-window correlation between domain-averaged MHWCI for Cluster 2 during summer and Interdecadal Pacific Oscillation (IPO) index using a fixed window length equal to that of the 2013-2021 period.

Figures using arrow vectors (particularly Figure 7) – Is it possible to make the arrows clearer as their direction is not easy to see?

**Response**: Thank you for this helpful suggestion. We will improve Figure 7 by reducing the vector density and increasing the arrow size to make the flow direction more easily discernible. In addition, we will simplify the vector representation by using arrow length to indicate current speed and a uniform color for all vectors, which will further enhance clarity and improve the readability of the figure.

Figure 7 – Colour bar for panels a and f uses a diverging colour scale and it would be better to use a non-diverging scale as the scale goes from 0 to 0.02.

**Response**: Thank you for this comment. We would like to clarify that the colour bars for panels (a) and (f) represent current speed, which is strictly non-negative, and a sequential scale from 0 to 0.02 m s$^{-1}$ is used. These colour bars are placed below panels (a) and (f) and are separate from the diverging colour scales used for other variables. To avoid confusion, we will revise the figure caption to more clearly indicate the range and meaning of the colour scale in panels (a) and (f): (a), (f) Geostrophic oceanic current (vectors, m/s, using a sequential colour scale from 0 to 0.02 m s$^{-1}$)

Figure 7 – Are the MOC and ZOC calculated at a particular latitude / longitude?

**Response**: Thank you for this question. The MOC and ZOC shown in Figure 7 are not calculated at a single fixed latitude or longitude. Instead, they are obtained by integrating the velocity field across the entire zonal or meridional extent of the study domain, respectively. Therefore, the resulting MOC and ZOC represent basin-integrated overturning circulations rather than transports along a particular section. We will clarify this in the revision.

Figure 7 convergence and overturning circulation colour scales - What direction do negative or positive numbers indicate?

**Response**: Thank you for pointing this out. We apologize for not clearly stating this in the original caption. In Figure 7, "negative stream function values correspond to clockwise circulation, while positive values indicate counterclockwise circulation". This information will be added to the figure caption for clarity.

Technical comments:

Line 400 – Should it say AMV instead of AMO?

**Response**: Thank you for pointing this out. You are correct. We will revise the manuscript and consistently use AMV.

Fig 2c and d – Series is misspelt in the titles.

**Response**: Thank you. The spelling error will be corrected in the revised figure.

Figure 7d, e, i, j – Suggest adding Latitude or Longitude labels on the x axis.

**Response**: Thank you for this helpful suggestion. Latitude/longitude labels will be added to the x axis in panels 7d, e, i, and j.

Figure 8 – budget is spelt wrong in the title.

**Response**: Thank you for pointing this out. The spelling error will be corrected in the revised figure.

Figure 9 – Colour bar should be labelled

**Response**: Thank you for pointing this out. The colour bar in Figure 9 will be labelled with the appropriate unit ($m^2\ s^{-2}$).

---

## Author Comment (AC3)

The manuscript examines the spatio-temporal variability of marine heatwaves (MHWs) in the North Sea and explores how different large-scale climate variabilities interact to shape their occurrence and characteristics. The authors use long-term SST observations, identify two main regional patterns of MHW variability, and relate these patterns to different atmospheric and oceanic climate modes, with a primary focus on seasonal differences.

Overall, the topic is important and highly relevant, especially given the increasing frequency and impacts of marine heatwaves in shelf seas. The focus on the North Sea is well chosen, and the approach of examining the combined influence of climate modes, rather than treating them independently, is interesting. The study appears scientifically sound, and the results are clearly relevant to the community.

However, I believe the manuscript still requires major revisions. My main concern the lack of several methodological details and the presentation, particularly in the results section, which makes it difficult to follow the main results and understand them. With significant improvements in clarity and structure, the paper could be much stronger.

**Response**: Thank you very much. We carefully considered each of your comments and suggestions, and we would like to respond to them in the following content.

**Introduction**
1. The introduction provides useful background, but the aim of the study is not clearly stated. I strongly recommend rewriting the objectives part of the introduction to clearly state the goal(s) of the work and the main research questions.

**Response**: We thank the reviewer for this constructive suggestion. We agree that the original Introduction does not sufficiently highlight the specific aims of the study. We will rewrite the objectives part of the Introduction to clearly state the main goals of the work and the specific research questions regarding the regional variability of marine heatwaves and their links to large-scale climate modes.

2. Additionally, since the analysis relies heavily on different climate modes, it would be helpful to define these climate modes early on, either in the introduction or in the data/methodology section. And add a brief explanation of how the positive and negative phases of these modes could affect SST variability in the North Sea.

**Response**: We agree that a clear definition of the climate modes and their relevance to North Sea SST variability should be provided early in the manuscript. We will revise the manuscript to explicitly define all climate modes used in this study and to clarify their physical relevance. Specifically, a new subsection (Sect. 2.4) will be added to the Methods section, which details the definitions, data sources, and calculation methods of the NAO, EAP, and AMV indices. In addition, the Introduction will be revised to include a concise description of how the positive and negative phases of these climate modes can influence SST variability and marine heatwaves in the North Sea through atmospheric circulation, and oceanic processes.

3. One minor point: the manuscript sometimes refers to the Atlantic Multidecadal Oscillation as AMO and sometimes as AMV. Please use only one term throughout the manuscript, as switching between them is confusing.

**Response**: Thank you for pointing this out. We will revise the manuscript and consistently use AMV.

**Data and Methods**
This section requires the most attention, primarily because important details are missing or not clearly described.

1. The data section is not sufficiently clear. I suggest rewriting it in a more straightforward manner, listing the datasets used, their sources, temporal and spatial resolution, and how they are used in the analysis. Currently, it is difficult to understand exactly which products are used and why.

**Response**: We agree that the current Data section is not sufficiently clear and that the description of the datasets and their roles in the analysis can be confusing. In the revised manuscript, we will rewrite and reorganize the Data section in a more straightforward and structured manner. Specifically, we will clearly list all datasets used, including their sources, temporal and spatial resolutions, and describe how each dataset is used in the different parts of the analysis. This reorganization will also incorporate additional references and details for key datasets such as ERA5 and ORAS5, addressing related comments raised by you and Reviewer 1.

2. The SST analysis is limited to 2021, but high-resolution SST products are available at least until the end of 2024. It is unclear why the study stops at 2021. If this limitation is due to other variables or datasets, it should be clearly stated. Otherwise, extending the analysis to include the most recent years would strengthen the study, especially since recent years include strong extremes that could affect the statistics.

**Response**: Thank you for raising this important point regarding the temporal coverage of the SST analysis. The original manuscript uses the OSTIA reprocessed SST product, which is a long-term, homogeneous dataset specifically designed for climate studies and marine heatwave statistics. At the time when the data were processed and the manuscript was prepared, this reprocessed dataset was available only up to 2021, which explains the end year of the original analysis.

We note that near-real-time OSTIA SST products, which extend to present day, are available. However, these products are not reprocessed and may involve changes in input data streams and processing procedures over time, making them less suitable for consistent long-term climatological and MHW analyses. For this reason, we did not combine the NRT product with the reprocessed dataset.

We realized that the reprocessed OSTIA SST dataset has been updated and now extends through 2023. In the revised manuscript, we will extend the SST analysis to include the newly available

years (2022-2023), assess their influence on the MHW statistics, and update the temporal coverage and rationale for dataset selection in the Data section.

3. Regarding the marine heatwave definition, the authors use the standard Hobday et al. method, but the climatology period used to define the MHW threshold is not mentioned. It must be included in the methodology. Also, please clarify whether the same climatology baseline period is used when computing other anomalies in the study, as this is not currently clear.

**Response**: The marine heatwaves are identified using the standard Hobday et al. methodology, with the climatological mean and threshold computed over the 1982-2021 period, consistent with the temporal coverage of the SST dataset. In addition, all other anomalies analyzed in this study are calculated relative to the same 1982-2021 climatological baseline. We will clarify these points in the Methods section.

4. Another confusing aspect is the detrending or trend removal. In lines 109–110, the manuscript refers to trended and detrended data, but it is not clearly explained: how the detrending is performed? and how it affects the results? This needs a clearer explanation, as detrending choices can influence both clustering and correlations with climate modes.

**Response**: The detrending procedure will be clarified in the revised manuscript. Specifically, long-term warming is removed by estimating a linear trend of the global-mean SST over the study period (1982-2021). Monthly global-mean SST is then calculated and linearly interpolated to daily resolution, after which a linear regression is applied to obtain the global SST trend. This global trend is then subtracted from the original SST field prior to marine heatwave detection.

This approach follows common practice in MHW studies and is intended to remove the externally forced warming signal while retaining internal variability. As a result, the clustering and the relationships with climate modes reflect variability relative to the evolving background state, rather than being dominated by the long-term warming trend. We will add a description of this procedure and its rationale in the Methods section.

5. I also recommend a minor change in the subsection title in the methods (line 106). The subtitle is "Cumulative Intensity of Marine Heatwaves," but the section actually describes the overall MHW calculation and detection method and includes multiple metrics. A title such as "Marine Heatwave Calculation" would be more accurate and clearer.

**Response**: We agree that the original subsection title was too specific. The subsection title will be updated to more accurately reflect the content of the section.

6. I was confused by the calculation of the "cumulative intensity anomaly." Since MHWs are already defined as anomalous events, it is not clear why an additional anomaly is needed. If the authors choose to keep this variable, they need to clearly explain its importance and what additional information it provides.

**Response**: Thank you for raising this point. The MHWCI anomaly is introduced for a different purpose. In this study, the cumulative intensity is used to quantify the total impact of all MHW events occurring within a given year or season. The anomaly of this metric is then calculated to emphasize interannual variability, allowing a direct comparison of how active or inactive different years are relative to the long-term mean. We acknowledge that this distinction was not sufficiently clear in the current manuscript. In the revised version, we will clarify the rationale for using cumulative intensity anomalies and explain the additional information they provide beyond the event-based MHW definition.

7. Since the analysis relies heavily on seasons, the seasonal divisions should be clearly stated and justified. Currently, there is no clear explanation of why these particular seasonal divisions are chosen or what they represent physically for the North Sea.

**Response**: Thank you for this comment. We acknowledge that, although the seasonal divisions (DJF, MAM, JJA, SON) are explicitly defined in the Methods section, their physical relevance to the North Sea was not sufficiently explained.

These seasonal divisions correspond to well-established physical regimes in the North Sea, including weakening of the winter thermal stratification and summer stratification with shallow mixed layers. Similar seasonal definitions have been widely adopted in previous studies on North Sea MHWs, which highlight pronounced contrasts between winter and summer mechanisms (Chen & Staneva, 2024; Mohamed et al., 2023). We will clarify this in the revision.

**Results**

For me, the Results section is currently difficult to follow, not because the figures are unclear, but because the text does not clearly describe the results shown in the figures. In many paragraphs, the text moves directly to interpretation and discussion without first explaining what is actually being observed. As a result, the reader must go back and forth between the figures and the text to understand what is being claimed.

1. I strongly recommend rewriting the Results section to be more descriptive: each figure should be clearly explained first (what pattern is shown, what changes occur, what differences appear), and only after that should the interpretation be introduced. This would make the manuscript much easier to read.

**Response**: We thank the reviewer for this constructive assessment. We agree that the current Results section places too much emphasis on interpretation and does not sufficiently describe the observed patterns shown in the figures.

In the revised manuscript, we will rewrite the Results section to provide a more readable structure. For each figure, we will first describe the key observed features (e.g., spatial patterns, temporal variability, and contrasts between regions or periods), and then physical interpretation and discussion. We believe that this reorganization will make the Results section more comprehensible and logically structured, while preserving the scientific content of the analysis.

2. Additionally, in lines 355-361, the manuscript separates the analysis into periods before and after 2013, but it is not clear why 2013 was chosen as a breakpoint. Where is this significant in the results? What evidence supports this split? This section needs more explanation.

**Response**: We thank the reviewer for requesting clarification on the rationale for choosing 2013 as a breakpoint. We would like to clarify that this choice is not arbitrary but is supported by both our results and independent evidence from existing literature.

First, our moving-window correlation analysis (Fig. R1) reveals a clear change in the relationship between summer MHWCI in Cluster 2 and the IPO around 2013, indicating a shift in the teleconnection structure at that time. We will add this figure to the supplementary of revised manuscript.

Second, previous studies have documented an abrupt cooling in the subpolar North Atlantic during 2013-2014, often referred to as the development of the "cold anomaly" or "cold blob," which coincides with a weakening of the AMV signal during the early 2010s (Frajka-Williams et al., 2017; Josey & Sinha, 2022; Moat et al., 2020).

In the revised manuscript, we will clarify this by explicitly linking the period separation to the moving-window correlation results and by citing relevant studies documenting the North Atlantic cooling and the contemporaneous weakening of the AMV. This additional explanation will make the physical motivation for the 2013 breakpoint clearer and better supported.

[Figure]

Figure R1. Sliding-window (8-year) correlation between domain-averaged MHWCI for Cluster 2 during summer and Interdecadal Pacific Oscillation (IPO) index.

**Minor Comments**

L26–27: The sentence is long and somewhat difficult to understand; consider splitting it for clarity.

**Response**: This sentence will be split to shorter sentences to improve clarity.

L106–112: Please clarify the detrending approach and its spatial application.

**Response**: We will clarify the detrending approach and its spatial application in the revised Methods. Specifically, detrending is performed by removing a linear trend estimated from the global-mean SST time series over 1982-2021. Monthly global-mean SST is first calculated and linearly interpolated to daily resolution, and a linear regression is then applied to estimate the long-term warming trend. This single global trend is then subtracted uniformly from the full SST field, rather than applying grid-point-wise detrending.

This approach removes the externally forced warming signal while preserving the spatial structure and internal variability of SST anomalies, thereby minimizing the influence of long-term trends on marine heatwave detection, clustering, and climate-mode relationships. These points will be clarified in the Methods section.

L118–124: Seasonal definitions should be explicitly stated.

**Response**: The seasonal definitions are explicitly stated in the Methods section (Lines 128-130), where seasons are defined as DJF, MAM, JJA, and SON, with DJF treated as a cross-year season.

L194: The sentence structure is heavy; consider rephrasing.

**Response**: We will rephrase this sentence to improve clarity and readability.

Reference used in the response

Chen, W., & Staneva, J. (2024). Characteristics and trends of marine heatwaves in the northwest European Shelf and the impacts on density stratification. *8th edition of the Copernicus Ocean State Report (OSR8)*, *4-osr8*, 7. https://doi.org/10.5194/sp-4-osr8-7-2024

Frajka-Williams, E., Beaulieu, C., & Duchez, A. (2017). Emerging negative Atlantic Multidecadal Oscillation index in spite of warm subtropics. *Scientific Reports*, *7*(1), 11224. https://doi.org/10.1038/s41598-017-11046-x

Josey, S. A., & Sinha, B. (2022). Subpolar Atlantic Ocean mixed layer heat content variability is increasingly driven by an active ocean. *Communications Earth & Environment*, *3*(1), 111. https://doi.org/10.1038/s43247-022-00433-6

Moat, B. I., Smeed, D. A., Frajka-Williams, E., Desbruyères, D. G., Beaulieu, C., Johns, W. E., Rayner, D., Sanchez-Franks, A., Baringer, M. O., Volkov, D., Jackson, L. C., & Bryden, H. L. (2020). Pending recovery in the strength of the meridional overturning circulation at 26° N. *Ocean Sci.*, *16*(4), 863-874. https://doi.org/10.5194/os-16-863-2020

Mohamed, B., Barth, A., & Alvera-Azcárate, A. (2023). Extreme marine heatwaves and cold-spells events in the Southern North Sea: classifications, patterns, and trends. *Frontiers in Marine Science*, *10*, Article 1258117. https://doi.org/10.3389/fmars.2023.1258117